# Adsorptive Membranes Incorporating Ionic Liquids (ILs), Deep Eutectic Solvents (DESs) or Graphene Oxide (GO) for Metal Salts Extraction from Aqueous Feed

**DOI:** 10.3390/membranes13110874

**Published:** 2023-11-03

**Authors:** Liyan Qalyoubi, Ioannis Zuburtikudis, Hadil Abu Khalifeh, Enas Nashef

**Affiliations:** 1Department of Chemical Engineering, Abu Dhabi University, Abu Dhabi P.O. Box 59911, United Arab Emirates; liyan.qalyoubi@adu.ac.ae (L.Q.); hadil.abukhalifeh@adu.ac.ae (H.A.K.); 2Department of Chemical Engineering, Khalifa University, Abu Dhabi P.O. Box 127788, United Arab Emirates; enas.nashef@ku.ac.ae

**Keywords:** desalination, membranes, graphene oxide (GO), ionic liquids (ILs), deep eutectic solvents (DESs), metal salts

## Abstract

Water scarcity is a significant concern, particularly in arid regions, due to the rapid growth in population, industrialization, and climate change. Seawater desalination has emerged as a conventional and reliable solution for obtaining potable water. However, conventional membrane-based seawater desalination has drawbacks, such as high energy consumption resulting from a high-pressure requirement, as well as operational challenges like membrane fouling and high costs. To overcome these limitations, it is crucial to enhance the performance of membranes by increasing their efficiency, selectivity, and reducing energy consumption and footprint. Adsorptive membranes, which integrate adsorption and membrane technologies, offer a promising approach to address the drawbacks of standalone membranes. By incorporating specific materials into the membrane matrix, composite membranes have demonstrated improved permeability, selectivity, and reduced pressure requirements, all while maintaining effective pollutant rejection. Researchers have explored different adsorbents, including emerging materials such as ionic liquids (ILs), deep eutectic solvents (DESs), and graphene oxide (GO), for embedding into membranes and utilizing them in various applications. This paper aims to discuss the existing challenges in the desalination process and focus on how these materials can help overcome these challenges. It will also provide a comprehensive review of studies that have reported the successful incorporation of ILs, DESs, and GO into membranes to fabricate adsorptive membranes for desalination. Additionally, the paper will highlight both the current and anticipated challenges in this field, as well as present prospects, and provide recommendations for further advancements.

## 1. Introduction

Water scarcity is a significant and escalating challenge for both humanity and the environment. The combination of population growth, industrialization, and climate change has placed immense pressure on our water resources [1]. This excessive exploitation of freshwater reserves not only jeopardizes food security, agriculture, and human well-being in numerous regions but also has global implications for water availability [2]. Projections indicate that by 2030, a mere 60% of the world’s water demand will be met, underscoring the urgency of the situation. Disturbingly, as of 2020, approximately 785 million people (1 in 9) lack access to clean water [3]. Moreover, it is predicted that by 2050, nearly 40% of the global population will reside in water-stressed areas, exacerbating the severity of the water scarcity issue [4]. Given these daunting challenges, there is an imperative to develop innovative and cost-effective approaches to water treatment.

Consequently, extensive research efforts have been dedicated to exploring wastewater treatment and the reutilization of water resources. The focus lies on the removal of diverse pollutants, such as heavy metal ions [5], dyes [6], and pharmaceutical compounds [7]. These studies represent important strides towards the creation of sustainable solutions, alleviating the strain on freshwater reserves and ensuring the availability of clean water for future generations.

Seawater desalination is a vital and highly effective water treatment technology that offers a reliable solution to the pressing issue of potable water scarcity in various regions [8]. Notably, approximately 97.5% of the global water supply consists of saltwater, with only around 2.5% being freshwater [9]. Desalination technologies are employed to extract salts and minerals from saline water, including substances such as NaCl, MgCl_2_, and CaCl_2_. These technologies can be categorized into three main groups: thermally activated systems that utilize evaporation and condensation processes, pressure-activated systems that apply pressure to drive clean water through semipermeable membranes, and chemically activated desalination techniques [9]. The widely used membrane technology, encompassing methods such as reverse osmosis (RO), forward osmosis (FO), membrane bioreactors (MBs), membrane distillation (MD), electrodialysis (ED), and nanofiltration (NF), plays a pivotal role in salt removal from water. The resulting desalinated water finds versatile applications, including providing safe drinking water, supporting irrigation needs, and meeting the requirements of the food industry [10].

The desalination process primarily relies on pressure-driven reverse osmosis (RO) membrane technology [11]. In RO applications, seawater is subjected to pressure and forced through a membrane, resulting in the separation of salt from the water as it moves from areas of higher concentration to areas of lower concentration [12]. Despite its efficiency and reliability, RO has several drawbacks, including high energy consumption, significant capital and operational costs, and issues such as membrane fouling, scaling, and high-pressure requirements [13]. In recent years, nanofiltration membranes have emerged as a promising alternative to address these limitations. These membranes operate at lower pressures, exhibit higher water flux rates, and effectively remove divalent salts and organic molecules, similar to RO membranes [14,15]. They have shown great potential for various applications, including water softening and the production of drinking water [16,17]. Consequently, substantial efforts have been dedicated to advancing desalination technology, with a focus on improving productivity, selectivity, and energy efficiency, and reducing the overall environmental impact [12]. These technological advancements are crucial for establishing desalination as a significant and sustainable source of urban potable water [18]. Therefore, it is essential to explore innovative approaches to salt removal that can further enhance the desalination process and ensure long-term water security.

Membrane engineering presents a highly effective solution to enhance the performance of existing membranes by incorporating cutting-edge materials. The ultimate goal is to achieve superior selectivity, permeability, mechanical strength, antifouling properties, and cost-effectiveness. One approach involves the development of nanocomposite or nano-enhanced membranes, where nanomaterials like metal oxides (such as zeolites and silica), carbon-based nanotubes, and hydrophilic functional groups are seamlessly integrated into the polymeric membrane matrix [19,20]. Furthermore, the integration of innovative materials like graphene-based compounds further elevates membrane performance. Recent advancements have witnessed the application of molecularly engineered frameworks, such as metal–organic frameworks (MOFs), covalent organic frameworks (COFs), two-dimensional (2D) materials (e.g., MXene), natural polyphenols (such as tannic acid and dopamine), and carbon nanoparticles like graphene oxide (GO), in the fabrication of nanofiltration membranes [14]. These composite membranes have demonstrated remarkable performance, achieving heightened permeability while operating at lower pressure requirements and maintaining outstanding salt rejection efficiency [12]. Moreover, ionic liquids (ILs) have gained significant recognition as environmentally friendly materials with tailored properties and exceptional adsorption capacity for water treatment purposes [21]. However, due to their cost implications, deep eutectic solvents (DESs) have emerged as cost-effective alternatives, offering similar desirable properties to ILs [22]. The primary objective of this study is to comprehensively explore the utilization of emerging materials and their integration into polymeric membranes to facilitate the extraction of salts from water.

## 2. Emerging Materials Utilized in Salts Extraction

### 2.1. Graphene Oxide (GO)

Graphene-based materials have gained significant acclaim in the scientific community for their vast range of promising applications across diverse fields such as engineering, electronics, biomedical sciences, energy, and many other scientific disciplines [23]. Graphene, an extraordinary two-dimensional carbon allotrope with a monoatomic thickness, possesses exceptional properties including high charge transport, thermal conductivity, optical characteristics, and mechanical strength [24]. Notably, graphene oxide (GO) stands out as a remarkable material with a distinct structure obtained through the exfoliation of graphite using the Hummer method [25,26,27]. GO is a highly advantageous precursor for obtaining graphene with enhanced yield and cost-effectiveness. It comprises a single-layered structure consisting of carbon, hydrogen, and oxygen molecules, making it readily available and economical [28]. It can also be utilized to produce ultrasmall nano-graphene oxide (NGO) [25]. The unique properties and versatile applications of GO stem from its monomolecular layer of graphite, adorned with various oxygen-containing functionalities such as epoxide, carbonyl, carboxyl, and hydroxyl groups [29]. Figure 1 illustrates the structure of GO, showcasing its distinctive features. The abundant availability and exceptional characteristics of GO position it as an attractive material for numerous industrial and scientific applications.

Graphene oxide (GO) stands out as a highly hydrophilic material, known for its excellent dispersibility in water and its biocompatibility, making it ideal for a wide range of applications [24,29]. GO can be employed in various forms, including nanoparticles and nanosheets, providing versatility in different configurations [31]. The incorporation of GO nanoparticles into membranes brings several advantages, such as improved hydrophilicity, enhanced mechanical strength, and increased thermal stability [32]. The functional groups present in graphene oxide facilitate its dispersion in polymeric solutions, leading to enhanced membrane performance, particularly in water purification applications [32]. This addition also contributes to reducing the pressure requirements during operation, resulting in more efficient water transport. Additionally, the integration of GO into membranes improves their resistance to fouling by enhancing hydrophilicity and reducing surface roughness [33]. Graphene oxide has attracted significant attention in the field of water treatment, particularly in desalination processes, owing to its exceptional chemical stability, superior mechanical strength, and excellent transport properties [34]. In a study, a GO-based supercapacitor was developed for arsenic removal and seawater desalination, demonstrating an impressive adsorption capacity for Na+ ions of 122 mg·g^−1^ [35]. Various fabrication techniques, such as dip-and-spin coatings, layer-by-layer assembly, vacuum filtration [35], and grafting [36], can be utilized to incorporate GO into membranes. These techniques offer flexibility and control, enabling the seamless integration of GO into membranes for diverse water treatment applications.

Moreover, the exceptional two-dimensional structure and tunable physicochemical properties of graphene oxide (GO) have led to the advancement of GO membranes by stacking nanosheets. These membranes possess remarkable characteristics such as ultra-thinness, high permeability, and energy efficiency, making them highly suitable for the selective sieving of ions and molecules in aqueous solutions [37]. However, when used as standalone membranes in water desalination, GO membranes tend to exhibit undesired swelling. This swelling phenomenon occurs as molecules traverse the graphene channels between the layers of graphene oxide, resulting in an expansion of the interlayer spacing. Consequently, larger salt ions can pass through the membrane while restricting the passage of smaller water molecules. To address this issue, the incorporation and dispersion of GO into polymeric membranes have emerged as promising strategies [38].

### 2.2. Ionic Liquids (ILs)

Ionic liquids have captured significant attention among researchers due to their unique properties and versatile range of applications. These substances, known as “green solvents”, have gained popularity as environmentally friendly alternatives to volatile organic solvents [39]. Consequently, they have been extensively investigated for various industrial and laboratory purposes. The term ionic liquids implies that the liquid is only composed of ions, with no molecular species present [40]. They typically comprise an organic cation and a polyatomic inorganic anion, which combine to form a salt in the liquid phase. They can exist as fused salts, molten salts, or liquid organic salts at or below room temperature, generally ≤100 °C [41]. To maintain a liquid state at room temperature, the cation of the ionic liquid typically possesses an asymmetrical structure with different alkyl groups [42]. Examples of commonly encountered cations and anions in ionic liquids include (CH_3_)_3_NHCl/2AlCl_3_, 1-butyl-3-methylimidazolium, tetrafluoroborate [BF_4_]^−^, 1-ethyl-3-methylimidazolium, and [PF_6_]^−^ [43], in addition to others presented in Figure 2.

A significant attribute of (aprotic) ionic liquids (ILs) is their extremely low vapor pressure, typically measured in pascals (Pa). This quality effectively resolves significant issues linked to volatile organic solvents, including ozone layer depletion, threats to human health, and solvent loss during processes due to their high volatility. This single characteristic alone has led to the recognition of ILs as “environmentally friendly solvents”, although it does not account for the various potential pathways through which ILs might enter the terrestrial ecosystem [45]. Additionally, ionic liquids (ILs) possess a wide range of attractive thermal, chemical, physical, and biological properties. They demonstrate characteristics such as low viscosity, negligible vapor pressure at room temperature, excellent thermal stability, versatile combinations of different ions, and adjustable solubility, acidity, and basicity. ILs also exhibit low corrosiveness, stability in both air and water, and immiscibility with many organic solvents. Moreover, they are renowned for being environmentally benign, nonflammable, nontoxic, and highly efficient in diverse applications. These ILs can be tailored to specific requirements by selecting and altering the cations and anions, enabling the design of “designer solvents” with tailored hydrophobicity, polarity, and solvent power [41,46]. Nevertheless, predicting IL properties and managing their inherent complexity can present certain challenges; therefore, generalizing their properties remains inaccurate [47]. Ionic liquids have shown great promise across a wide range of disciplines, including catalysis [48], polymer dissolution and transformation [49], physical chemistry [50], polymer science [51], cell biology [52], material science [53], electrochemistry [54], desalination [55], energy production [56], nuclear physics [57], medicinal chemistry [58], and various engineering fields [41]. For instance, one potential strategy relies on the inclusion of metal complexes as part of ionic liquids (ILs) as they aid the solubilization of metal complexes in nonaqueous environments. In one study [59], two series of redox active metal–EDTA complex-based ILs have been prepared and inspected for the solubility analysis. The findings suggest that [TOMP] series ionic liquids (ILs) offer a proficient approach to efficiently dissolve metal–EDTA complexes in nonaqueous solutions. The outstanding solubility features, favorable thermal performance, and impressive electrochemical properties showcased by the investigated reversible redox couples render the ILs viable for potential employment in electrochemical applications. These applications hold relevance in both aqueous and nonaqueous environments [59].

As the research on ionic liquids has progressed, different categories have emerged to cater to specific applications and properties. Among these categories are room-temperature ionic liquids (RTILs), which are utilized for separating carbon dioxide (CO_2_) from hydrogen sulfide (H_2_S) during the sweetening process of natural gas [60]. Task-specific ionic liquids (TSILs) offer customization options by incorporating specific functionalities, such as metal-ligating groups, to enhance the extraction of desired metal ions from water [61]. Polymerized ionic liquids (PILs), also known as polyelectrolytes, consist of repeating units that contain ionic liquid species, and they find applications in separation, absorption, and catalysis [62]. Supported IL membranes (SILMs) are composite materials where ILs are supported on different substrates like metal–organic frameworks, which serve as efficient adsorbents in various applications [63].

Furthermore, ionic liquids can be classified into two main categories: simple salts and binary ionic liquids. Simple salts consist of a single anion and cation, while binary ionic liquids are composed of mixtures involving an equilibrium. For instance, [EtNH_3_][NO_3_] represents a simple salt, whereas mixtures containing aluminum(III) chloride and 1,3-dialkylimidazolium chlorides represent a binary ionic liquid system with multiple ionic groups. The properties and melting points of binary ionic liquids are influenced by the mole fractions of the different ionic species present in the mixture [42].

Ionic liquids (ILs) have gained significant attention in the field of water treatment due to their diverse range of applications. The focus of this paper is specifically their potential for desalination, with previous studies exploring their effectiveness in removing metal salts from water and investigating their solubility properties. For example, one study found that at a temperature of 125 °C, NaCl exhibited a solubility of 8.96 wt% in 1-ethyl-3-methylimidazolium dimethyl phosphate. Notably, imidazolium-based ILs demonstrated higher solubility for NaCl compared to pyridinium-, pyrrolidinium-, and ammonium-based ILs [64]. Furthermore, the IL [emim][Tf_2_N] has been successfully employed in directional solvent extraction (DSE) for desalinating saltwater, offering high production rates and low energy consumption [55]. This DSE is not water-soluble but can effectively reject salt ions [55]. Various studies investigating the solubility of salts in ILs are summarized in Table 1.

Despite the fact that ionic liquids are considered biocompatible solvents due to their nonvolatile nature, toxicological studies proved that the belief of their nontoxicity as a result of that is incorrect. Overall, the findings revealed that ionic liquids (ILs) exhibit fungicidal, bactericidal, or herbicidal activities, potentially acting as toxic agents when introduced into the environment. The degree of toxicity varies based on the ILs’ molecular structure and the specific organisms and conditions employed for toxicity assessment [68]. Numerous researchers have undertaken efforts to assess the toxicological impacts of ILs on various biological entities, including enzymes, mammalian cells, bacteria, microalgae, vertebrates, and invertebrates. Given the customizable nature of ILs, which permits the creation of diverse molecular structures, it is more appropriate to conduct empirical analyses of their harmful effects using the principle of structure–activity relationship (SAR). In these experimental trials, modifications were introduced to IL structures by altering factors such as the alkyl chain lengths in cations and anions, the types of head groups in cations, variations in anion types, and the arrangement of substituents. The observed adverse effects of ILs on bacteria were notably associated with the cationic component, particularly concerning the lengths of alkyl chains and the corresponding head groups. Cations exerted a more pronounced influence due to their heightened affinity for lipid solubility and their ability to interact attractively with microorganism phospholipid membranes through the electrostatic attraction of the positive charge. Several computational findings have revealed that the extended alkyl chain present in cations has the capacity to integrate itself within the lipid layer of the cellular membrane. Consequently, the cationic components establish interactions with the negatively charged phosphocholine or phosphoserine molecules within the cell membrane. In contrast, anions possess relatively smaller molecular structures and carry a negative charge, resulting in limited lipophilicity and a weaker propensity to adhere to microorganisms. While their impact on overall toxicity may not be as pronounced as that of cations, the specific choice in anions remains pivotal in modulating toxicity outcomes [68]. In conclusion, ionic liquids’ drawbacks related to toxicity can be avoided with the cautious choice in the constituents. 

### 2.3. Deep Eutectic Solvents (DESs)

Deep eutectic solvents (DESs) represent a promising alternative to ionic liquids and are considered the next generation of sustainable solvents. They possess unique properties that address the drawbacks of ILs, including issues related to toxicity, high cost, challenging preparation, purification requirements for reuse, and nonbiodegradability [69]. DESs represent a novel class of solvents that possess distinct properties compared to other types of solvents, primarily influenced by the characteristics of their constituent compounds. Therefore, by making wise choices in selecting initial compounds, DESs offer a promising and supplementary pathway in the quest for environmentally friendly solvent alternatives to replace traditional VOCs in specific applications [45]. Moreover, they share similar physicochemical characteristics with ILs but offer several advantages. These include easy accessibility, simple preparation through the mixing of natural components, and potential scalability. These solvents consist of hydrogen-bonding acceptors (HBAs) and hydrogen-bonding donors (HBDs), resulting in the formation of liquid eutectic solvents [70].

DESs are composed of eutectic mixtures of pure components for which the eutectic point temperature should be lower to that of an ideal liquid mixture [71]. They are often described as being natural, safe, and cost-effective [72]; however, those generalizations are inaccurate since these characteristics depend on the starting materials used and their mixture. Therefore, thorough investigations need to be conducted prior to their usage in any application [72]. These common generalizations often center around concerns about the toxicity and volatility of deep eutectic solvents (DESs), often without paying proper attention to the constituents that compose DESs. Similar to ionic liquids (ILs), the characteristics of DESs are intricately linked to the properties of the substances they comprise. It is crucial to note that DESs differ from ILs in that they are not pure fluids. Consequently, the properties of DESs are also influenced by their composition, specifically the ratio of hydrogen bond acceptors (HBA) to hydrogen bond donors (HBD). Thus, the components of DESs can have varying impacts on each property. Conversely, as nonideal mixtures, DESs might exhibit properties that do not fall simply within the range of those of their parent compounds. The extent of this deviation depends on the dominant interactions occurring within the bulk or at the surface, contingent upon the particular property being considered [45]. 

The preparation of DESs involves a straightforward process of mixing the constituents at low temperatures with magnetic stirring, offering energy- and time-saving benefits compared to ILs. Therefore, DESs are recognized as the green solvents of the 21st century with customizable solvent properties serving as viable alternatives to ILs [73]. On the other hand, being cautious with the choice in DESs for certain applications is essential, as contradictory results regarding their toxicity are arising. Some studies have shown that certain combinations of DES constituents can show a toxic or nonbiodegradable nature which can limit their potential uses at a large scale to evade environmental pollution [74]. However, these studies are very limited and contradictory, as some studies have shown no toxicological effects of some DESs on bacteria. For instance, a choline chloride (ChCl)-based DES was tested with four hydrogen bond donors including glycerine, ethylene glycol, triethylene glycol, and urea, which proved its benign effects on bacteria [75].

Common components utilized in DESs include urea, carboxylic acids, glycerols, and quaternary ammonium salts. Among these, choline chloride stands out as a notable and frequently employed cation due to its cost-effectiveness, biodegradability, and nontoxic nature as a quaternary ammonium salt [76]. Examples of common HBAs and HBDs can be found in Figure 3.

Deep eutectic solvents (DESs) are prepared through the formation of hydrogen bonds between their constituents, resulting in a eutectic mixture that remains liquid within the temperature range of room temperature to 70 °C [78]. While DESs share some similarities with ionic liquids (ILs), they are distinct in that DESs are not exclusively composed of ionic species and can be derived from nonionic components [78]. DESs possess unique properties, including high solubility for pollutants, a low melting point, high conductivity, inertness to water, availability of cost-effective raw materials, and ease of preparation [79]. These properties make DESs suitable for various applications such as parts cleaning, electropolishing, semiconductor processing, fuel desulfurization, metal processing, removal of mercury from gas streams [76], electrochemistry [80], metal deposition [80], and reaction solvents in radical-mediated hydrothiolation reactions [81]. Moreover, they serve as chiral mediums for enantioselective electroanalysis and are employed in the dissolution and extraction of the most prevalent polysaccharides from natural sources such as cellulose, chitin, starch, hemicelluloses, and pectin [82,83]. This is in addition to the extraction and recovery of polyphenols from agrifood byproducts [84]. DESs have also played a pivotal role in the pharmaceutical industry. They have not only improved the separation of actives from natural products but have also significantly enhanced the solubility and permeability of drugs. These enhancements are critical for drug absorption and efficacy [85]. Furthermore, DESs show significant potential in the separation of CO_2_, SO_2_, and H_2_S, and they have recently gained attention for their ability to absorb NH_3_ [86,87,88,89]. It is worth noting that the potential combinations of cations and anions for creating ionic liquids (ILs) are nearly limitless and the number of new compound mixtures leading to deep eutectic solvent (DES) systems is continually on the rise, including the recent discovery of hydrophobic DESs. As a result, it is only natural to expect that certain systems will be more appropriate for specific applications, while others will be better suited for different purposes [45].

The potential of deep eutectic solvents (DESs) for salt removal stems from their impressive solubility characteristics. A recent study aimed to investigate the solubility of sodium chloride (NaCl), sodium bromide (NaBr), and sodium carbonate (Na_2_CO_3_) in nine DESs based on ammonium compounds. The results revealed that N,N-diethylethanolammonium chloride:zinc(II) chloride exhibited the highest solubility for NaCl, with solubility levels increasing as the temperature escalated. Interestingly, DESs incorporating ethylene glycol or glycerol displayed relatively lower solubility, whereas those incorporating ammonium salts and metal halides demonstrated higher solubility [90]. These findings provide encouraging insights for the potential utilization of DESs incorporating ammonium compounds and metal halides in desalination membrane applications.

## 3. Adsorptive Membranes Technology with Emerging Materials

The increasing global demand for freshwater has spurred a search for sustainable desalination techniques. Among these, membrane-based desalination stands out as the most common, energy-efficient, and space-saving option. These compact and lightweight membranes offer high productivity, making them a preferred choice in the desalination industry [91]. However, the cost of desalination remains a major challenge, preventing many countries from adopting it as a viable water source. To tackle this issue, efforts are focused on developing innovative approaches and technologies that can enhance performance and reduce costs. A promising solution lies in the development of adsorptive membranes, which incorporate advanced materials into their matrix. These membranes hold great potential for revolutionizing the desalination process, improving efficiency, and making it more affordable, thereby addressing the global water scarcity issue.

Adsorptive membranes offer a powerful combination of adsorption and membrane technologies for effective pollutant removal from water sources [92]. These membranes present numerous advantages that address the limitations of typical water treatment methods, including fouling, high operating pressures, costly regeneration processes, and limited selectivity [93]. The purification process schematic representation of adsorptive membranes is shown in Figure 4. Referred to as modified membranes, they have demonstrated exceptional capabilities in efficiently eliminating various pollutants while maintaining high permeability, flux, recyclability, compactness, and space-saving features [94]. The fabrication process involves incorporating powders, nanomaterials, or flakes into a polymeric membrane support [93]. 

Polymeric membranes are widely used in conjunction with microfiltration, ultrafiltration, reverse osmosis, or nanofiltration techniques to produce clean water [96]. The phase inversion technique, commonly employed in membrane manufacturing, utilizes polymers that can dissolve in a solvent and have a miscible nonsolvent [97]. These membranes facilitate the advancements in materials, coatings, and manufacturing techniques, attracting researchers to explore diverse types of adsorbents, including nanosized adsorbents and bioadsorbents [96]. Consequently, the integration of emerging materials such as ILs, DESs, and GO into polymeric membranes holds significant promise for enhancing desalination processes [90]. The following sections will discuss the studies that have investigated the use of adsorptive membranes incorporating these materials for the removal of metal salts from their aqueous solutions.

### 3.1. Incorporation of Graphene Oxide in Membranes

Graphene-based materials have emerged as highly promising candidates for water treatment applications due to their multifunctional properties as photocatalysts, adsorbents, and disinfectants. These materials have demonstrated remarkable effectiveness in combating various pollutants such as organic molecules, heavy metals, and waterborne pathogens [98]. Among them, graphene oxide (GO) has garnered significant attention and is widely recognized for its exceptional adsorption capabilities. By combining GO with ionic liquids, it has been employed as a potent adsorbent for the extraction of dyes like reactive blue 19 (RB19) [99], as well as pharmaceutical compounds including sulfamethoxazole (SMZ), carbamazepine (CBZ), and ketoprofen (KET) [100]. The utilization of GO in water treatment is attributed to its large surface area, functional groups, amphiphilic nature, and versatile chemistry [101]. Abundant oxygen functional groups, such as hydroxyl, epoxy, and carboxyl, enable strong interactions with water molecules, enhancing their hydrophilicity [102,103,104]. Therefore, incorporating GO into adsorptive membranes offers a promising approach to enhance their performance. These GO-incorporated membranes exhibit improved permeation flux and enhanced resistance to fouling compared to conventional commercial membranes [105]. Moreover, the use of GO nanoparticles in modified membranes has shown exceptional efficacy in treating challenging water sources with high salinity, oil contamination, surfactants, and emulsions, making it a popular choice for selective water permeation [105,106].

Graphene oxide (GO) holds immense promise in the field of membrane desalination and can be incorporated into membranes using various approaches, including freestanding GO membranes, GO surface-modified membranes, and casted GO-incorporated membranes. The utilization of nanoporous freestanding graphene membranes has showcased exceptional efficacy in salt filtration, surpassing conventional reverse osmosis (RO) membranes by achieving significantly higher rates of NaCl removal from an aqueous solution of water [107]. Furthermore, GO has exhibited remarkable antibacterial activity, particularly against *Escherichia coli*, rendering GO membranes highly resistant to fouling [108]. In a different study, researchers successfully fabricated a freestanding nanofiltration membrane using GO via a drop-casting method. This membrane effectively separated sodium salt from organic pollutants and copper salt from the aqueous solution prepared, showcasing its potential for efficient water purification [109]. Molecular dynamics simulations (MDS) have further substantiated the exceptional performance of GO membranes in desalination processes. These simulations have revealed that GO membranes achieve the complete rejection of salt particles and possess significantly enhanced water permeability compared to conventional RO techniques, owing to their ultrathin structure [110]. 

Moreover, graphene oxide (GO) finds application in the modification of membrane surfaces, leading to improved characteristics such as enhanced antibacterial and chlorine resistance [111]. The utilization of GO nanomaterials in this process offers cost-effectiveness as well [112]. In another study, a sulfonated poly(arylene ether sulfone) reverse osmosis (RO) membrane underwent surface modification using a layer-by-layer deposition technique with GO and aminated graphene oxide (aGO) nanoparticles [113]. Figure 5 shows the cross-section SEM image of (GO/aGO) multilayers prepared on a PS support layer. The resulting modified membrane exhibited superior chlorine resistance compared to the pristine membrane and showcased enhanced performance, including an impressive NaCl salt rejection rate of 98%, a water flux of 28 L/m^2^h, and a low contact angle [113]. These tests were performed using a salt aqueous solution in a cross-flow cell instrument. 

Moreover, there have been numerous studies conducted on the incorporation of graphene oxide (GO) into polymer matrices to enhance membrane performance. One such investigation focused on modifying polysulfone polymeric membranes by adding varying amounts of GO (ranging from 0 to 0.5 wt.%) for the purpose of removing NaCl, MgCl_2_, Na_2_SO_4_, and MgSO_4_ from saltwater [114]. The results demonstrated that the inclusion of 0.3 wt.% GO yielded the highest water permeability, with rejection rates of 95.2% for Na_2_SO_4_, 91.1% for MgSO_4_, 62.1% for MgCl_2_, and 59.5% for NaCl. Furthermore, the modified membrane exhibited superior pure water flux, permeability, and salt rejection (up to 88.5%) compared to the pristine membrane. Additionally, the presence of GO contributed to increased hydrophilicity, surface negativity, and improved resistance against fouling [114]. Furthermore, these modified membranes not only exhibited enhanced performance but also demonstrated improved membrane characteristics. For instance, a polyethersulfone (PES) membrane incorporating a blend of graphene oxide in its matrix, followed by modification with hyperbranched polyethylenimine (HPEI), showcased higher tensile strength and Young’s modulus [115]. Additionally, it displayed enhanced hydrophilicity, increased surface negative charge, smoother surface topography, and improved antifouling properties against *Escherichia coli* [115]. In Table 2, other investigations that involve graphene oxide utilization in desalination are overviewed along with their performance.

Furthermore, recent studies have explored the use of functionalized graphene oxide (GO) to modify ceramic ultrafiltration membranes as an alternative to conventional polymeric membranes [36]. The introduction of GO led to significant improvements in the modified membrane’s hydrophilicity and increased the negative charge on its surface compared to the pure membrane. The GO coating played a crucial role in facilitating the efficient removal of various contaminants found in water, including natural organic matter, pharmaceuticals such as ibuprofen and sulfamethoxazole, and inorganic salts including NaCl, Na_2_SO_4_, CaCl_2_, and CaSO_4_. However, it is important to note that the removal efficiency for inorganic salts was relatively lower, with a minimum value of 31.4% achieved. These findings emphasize that polymeric membranes remain superior choices for the effective and cost-efficient removal of salts, owing to their higher removal efficiency, lower production costs, and ease of manufacturing [36].

### 3.2. Incorporation of Ionic Liquids in Membranes

Various strategies have been proposed to enhance the performance of conventional desalination membranes. One approach involves modifying the membranes using different materials, such as titanate nanotubes (TNTs) [124], zeolites [125], chitosan [126], hydrofluoric acid, and nitric acid [127]. However, some of these materials have drawbacks, including high volatility or toxicity, which can raise environmental concerns. On the other hand, ionic liquids (ILs) offer advantages such as nonflammability and low volatility. However, ILs also have limitations, including nonbiodegradability, high cost, difficulty in recycling, and potential toxicity. These drawbacks can be avoided through the proper selection of the IL constituents which can be decided via computational modeling studies joint with life cycle assessment studies to produce combinations that adhere with green chemistry principles [128]. Another promising solution to overcome these limitations is integrating ILs with membrane technology. This integration aims to improve the properties of ILs, enhance membrane performance, and mitigate their drawbacks. For instance, ILs can reduce the required duration of the IL phase for specific processes, improve the recovery and reusability of ILs, and enhance polymer–filler adhesion or thinning of the polymer active layer in membranes. This results in reduced resistance to molecular movement [129]. Nevertheless, applying ILs for the surface modification of thin-film composite (TFC) membranes remains a challenging task [130]. Further research is needed to overcome the obstacles and fully harness the potential benefits of ILs in enhancing membrane performance for desalination applications.

Ionic liquids (ILs) are regarded as environmentally friendly solvents that establish strong hydrogen bonds with polymers [75]. The interaction between ILs and polymers is influenced by the degree of ionization of the IL, which is temperature-dependent. Higher temperatures enhance the ability of ILs to dissolve polymers [131]. ILs serve various functions in membrane preparation. They can be used as raw materials for fabricating different types of membranes, such as bulk IL membranes (BILMs), emulsion IL membranes (EILMs), supported IL membranes (SILMs), and poly(ionic liquid) membranes (PILMs) [132]. Additionally, ILs can be employed as physical additives or chemical modifiers to improve membrane properties. SILMs are prepared by immersing the membrane support in ILs, with commonly used supports including polysulfone (PSF), polyvinylidene fluoride (PVDF), polyimide (PI), ceramic, and nanofiltration membranes [133]. On the other hand, PILMs are composed of polymerizable IL monomers and are primarily used for CO_2_ separation [134]. 

Membrane technology offers a widely adopted approach for enhancing the performance of polymeric membranes by incorporating ionic liquids (ILs). This integration of ILs can lead to material and cost savings, achieved through physical or chemical modifications. In a notable example of chemical modification, a polyamide thin-film composite membrane was subjected to surface modification using an imidazolium IL called 1-aminoethyl-3-methylimidazolium bromide (AMIB) [135]. The schematic representation of surface modification is shown in Figure 6.

The resulting modified membrane exhibited exceptional properties, including high water permeability, efficient salt rejection, and effective separation of antibiotics. Moreover, it displayed antibacterial activity with a remarkable 99% reduction in E. coli colonies. The adsorptive membrane outperformed the unmodified membrane by achieving a 95% rejection rate for Na_2_SO_4_ and demonstrating a fourfold increase in permeate. Additionally, it showcased excellent performance in separating antibiotic/NaCl salt mixtures, presenting significant advantages [135]. 

In research conducted by Zhang et al. [136], the desalination performance of a conventional reverse osmosis membrane was significantly improved through the incorporation of an ionic liquid (IL). The specific IL utilized in the study was 1,3-dimethylimidazolium dimethyl phosphate ([MMIM][DMP]), which was formed using an addition reaction. The membrane modification process involved immersing the commercial membrane in the IL for a period of 2 to 8 min using an activation method based on the temperature-dependent solubility of the ILs [137]. The results of the study demonstrated remarkable enhancements in the modified membrane’s properties. The active layer of the membrane became thinner, smoother, and more hydrophilic, with the optimal modification achieved at a temperature of 110 °C. These modifications led to notable improvements in anti-fouling properties, as indicated by a water flux recovery rate of 95.5%, a 62% increase in water flux, and a NaCl rejection rate of 93.2% [136].

Moreover, a recent research study focused on enhancing the performance of a polyamide thin-film composite reverse osmosis (TFC RO) membrane through the utilization of three distinct ILs: 1-butyl-3-methylimidazolium chloride ([BMIM][Cl]), 1-methyl-3-octylimidazolium chloride ([C8MIM][Cl]), and 1-butyl-3-methylimidazolium bromide ([BMIM][Br]) [138]. The modification process involved immersing the membrane in a solution containing these ionic liquids. The results of the study demonstrated notable improvements in membrane performance, including a 6.5% increase in MgSO_4_ rejection and a remarkable 87% enhancement in water flux. Moreover, the treated membranes exhibited decreased contact angles, indicating improved hydrophilicity, as well as reduced surface roughness, which in turn contributed to a reduced susceptibility to fouling. Importantly, the incorporation of the ionic liquids induced chemical structural modifications in the membrane, further enhancing its properties. Furthermore, the [BMIM]Cl ionic liquid was also employed to modify a commercially available polyamide RO membrane. The modification process resulted in the production of a thinner and more hydrophilic membrane. Remarkably, this modified membrane exhibited a substantial 20% increase in water flux compared to the pristine membrane. Additionally, the modification led to a decrease in surface roughness, thereby significantly improving the membrane’s antifouling properties [131].

In another research endeavor, the same ILs were used as well to enhance a polyamide thin-film composite reverse osmosis (TFC RO) membrane [139]. This innovative investigation involved the incorporation of polymeric ionic liquids (PILs) into porous carbon membranes, resulting in the creation of charged porous membranes with customizable pore structures [139]. By carefully selecting suitable anions within the PILs, the researchers successfully synthesized hierarchically porous carbon membranes. The resulting photothermal membrane exhibited highly promising performance in the realm of seawater desalination. However, it is worth noting that the utilization of PIL-based membranes for liquid separations remains an area with relatively limited research and exploration [139].

### 3.3. Incorporation of Deep Eutectic Solvents in Membranes

Deep eutectic solvents (DESs) have evolved as superior alternatives to conventional organic solvents and ionic liquids due to their adjustable properties, affordability, ease of preparation, biodegradability, and safety profile. These characteristics have propelled DESs as highly attractive solvents for a wide range of applications [140]. Toxicity investigations have demonstrated that DESs composed entirely of renewable biomaterials, such as urea, choline chloride, and water, exhibit nontoxic and environmentally benign properties [75]. Recent research has proposed a novel approach by incorporating DESs into membranes. In a notable study, biodegradable and cost-effective DESs, namely N-methylacetamide-acetamide, N-methyl acetamide-N-methyl urea, and N-methyl acetamide-N,N′-dimethyl urea, were prepared and employed to dissolve a polyvinylidene fluoride (PVDF) polymer. This integration led to the modification of PVDF membranes with enhanced characteristics, including improved water permeability and increased rejection of bovine serum albumin (BSA) [141]. However, it is important to note that the exclusive use of DESs for membrane modification in desalination applications has not been explored yet. This can be a topic for future research since these membranes have shown their potential in other applications. Careful selection of DESs is crucial, as their cost, toxicity, and biodegradability depend on the specific combination of hydrogen donor and acceptor. By making informed choices, DESs with competitive performance can be prepared for effective membrane fabrication [141].

### 3.4. Hybrid Membranes

The utilization of adsorptive membranes incorporating emerging materials holds great promise for water desalination. Among these materials, ionic liquids, deep eutectic solvents, and graphene oxide have demonstrated their competence as adsorbents in polymeric membranes. However, the combination of these materials to create hybrid membranes has received limited research attention. Hybrid membranes, which integrate multiple materials, have the potential to improve the filtration process and enhance the durability of membranes in water treatment applications [142]. In a recent study conducted by Liu et al., a novel approach was employed to modify graphene oxide (GO) using an ionic liquid and polydopamine. These modified nanosheets were then assembled with polyelectrolytes on a polyethersulfone (PES) membrane support, resulting in the creation of a composite GO membrane. The membrane was evaluated for its performance in dye/salt separation and yielded promising outcomes. Notably, it operated effectively at a remarkably low operational pressure of 0.5 bar, while maintaining a high permeability of approximately 38.4 LMH/bar (at a concentration of 100 ppm direct red 80 and 5 g/L NaCl). The incorporation of the ionic liquid modification also facilitated the efficient removal of dye molecules during alkaline membrane washing processes, thereby enhancing the overall effectiveness of the membrane [143]. Table 3 shows overviews of other studies that combined emerging materials to fabricate adsorptive membranes.

## 4. Challenges and Future Outlooks

The utilization of adsorptive membranes incorporating emerging materials such as ILs, DESs, and GO for the removal of metal salts presents a promising technology. Previous studies have demonstrated their competence in enhancing performance, reducing costs, and minimizing the environmental impact compared to conventional membranes. However, there are several challenges that must be addressed to successfully implement this technology. These challenges encompass the following:Understanding the role and mechanisms of these materials and their interactions with membranes: Further research is needed to enhance membrane separation performance by investigating possible downsides such as mechanical instability, durability, nonideal alignment and assembly, surface defects, toxicity, high-pressure resistance, leaching, and deterioration [112].Scaling up the fabrication process for commercial production: The ultrathin permeable membranes studied in laboratory environments need to be scaled up for industrial applications. Real-life experiments are necessary to examine different aspects of industrial-scale applications, particularly in synthesizing MMMs and nanocomposite membranes in configurations such as hollow fiber membranes, which provide greater packing density than flat-sheet membranes [146].Understanding material–water interactions and conducting toxicity analyses: Some fillers used in these membranes may possess toxic properties, and their application in water purification must be carefully evaluated. ILs and DESs, being diverse classes of materials with various combinations, pose challenges in fully understanding their interactions in water applications.Developing predictive models for adsorptive membranes: In-depth research is required to identify the morphology and distinct properties of these membranes. Predictive models need to be developed to anticipate membrane performance, including kinetics and selectivity [147].Addressing membrane fouling issues: Increasing the hydrophilicity of membranes can sometimes lead to membrane overload. Although functionalizing polymeric membranes with nanomaterials can reduce biofouling [148], excessive particle loading can result in the formation of macropores and agglomeration, which negatively impact ultrafiltration performance due to particle aggregation [149]. Therefore, careful consideration of the compatibility between the chosen adsorbent and membrane, as well as conducting fouling studies, is essential to minimize fouling.Selecting suitable DESs for optimal desalination performance: Since limited research has been conducted on utilizing DESs for membrane modification in desalination, the wide range of possibilities can make it challenging to choose the most appropriate material. Careful evaluation is necessary to determine the DESs that can achieve the best desalination performance.

## 5. Conclusions

In conclusion, the application of emerging materials in adsorptive membrane fabrication for water desalination presents a promising approach which encompasses the advantages of adsorption and membrane filtration technologies. Materials such as graphene oxide, ionic liquids, and deep eutectic solvents possess unique properties that have demonstrated their efficacy in removing metal salts. When incorporated into polymeric membranes, these materials have exhibited excellent performance and enhanced membrane characteristics. Numerous studies have reported increased pure water flux, permeability, rejection rates, and selectivity, while effectively mitigating membrane fouling issues. However, the implementation of adsorptive membranes with emerging materials faces several challenges that must be addressed to fully harness their potential. These challenges include ensuring the safety of these materials in terms of human health and environmental impact, scaling up the fabrication process for industrial applications, and accurately predicting the membrane’s performance when incorporating these materials. It is crucial to carefully select the appropriate components for ionic liquids and deep eutectic solvents, as well as gain a comprehensive understanding of their behavior within membrane systems. Future research efforts should focus on conquering these challenges to develop efficient adsorptive membranes with emerging materials for desalination purposes. By doing so, we can unlock the full capabilities of these materials and advance water desalination technologies to effectively tackle the global issue of water scarcity. 

## Figures and Tables

**Figure 1 membranes-13-00874-f001:**
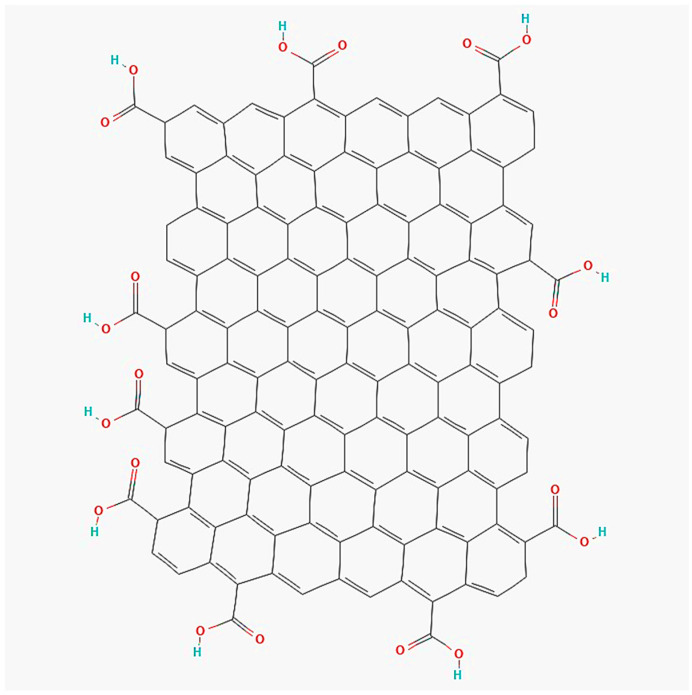
Graphene oxide structure [30].

**Figure 2 membranes-13-00874-f002:**
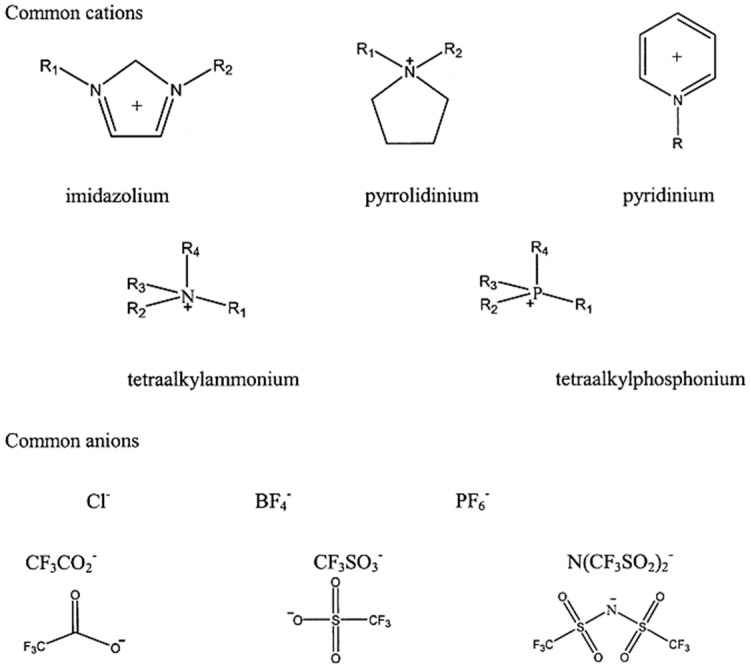
Structures of common cations and anions of ionic liquids [44].

**Figure 3 membranes-13-00874-f003:**
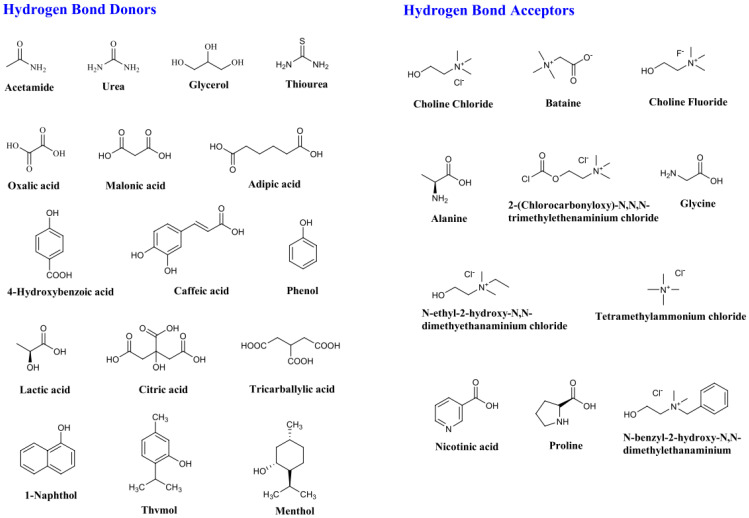
Chemical structures of common HBAs and HBDs [77].

**Figure 4 membranes-13-00874-f004:**
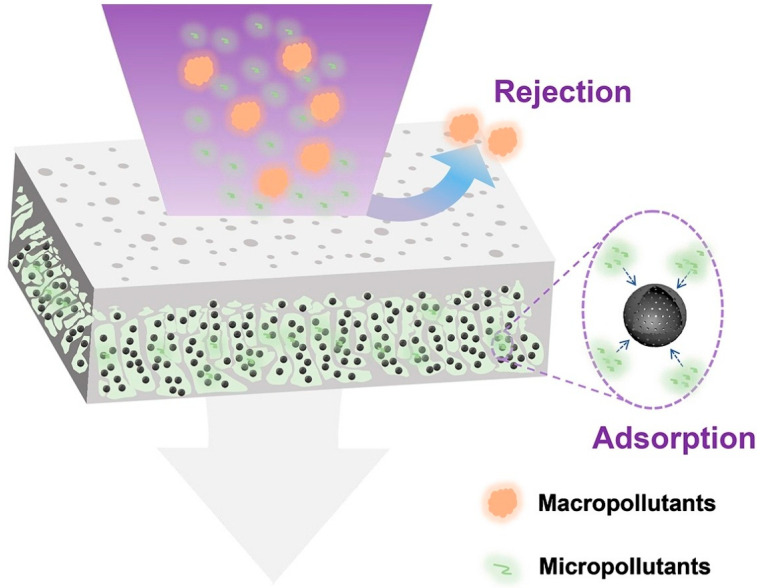
The purification process schematic representation of adsorptive membranes [95].

**Figure 5 membranes-13-00874-f005:**
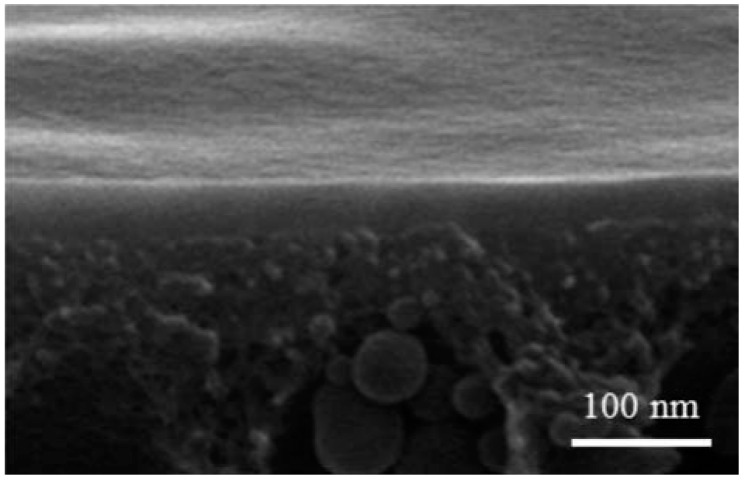
Cross-section SEM image of (GO/aGO) multilayers on PS support layer [113].

**Figure 6 membranes-13-00874-f006:**
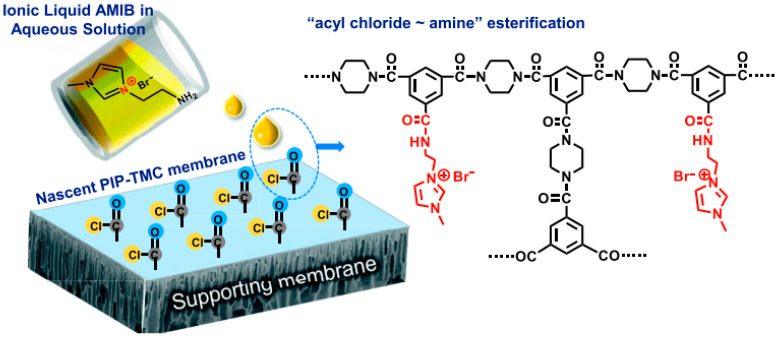
Surface modification of polyamide TFC membranes by AMIB IL [135].

**Table 1 membranes-13-00874-t001:** Studies regarding the solubility of metal salts in ILs.

Solubility of Metal Salts in ILs	Main Findings	Reference
NaCl, MgCl_2_, and CaCl_2_ solubility in 1,3-Dimethylimidazolium dimethyl phosphate was studied	Highest solubility achieved is with MgCl_2_ with a maximum concentration of 15,115 ppm at 100 °C. In contrast, it was 6639 ppm for NaCl at 60 °C and 1393 ppm for CaCl_2_ at 25 °C.	[65]
Hydrophobic tetraalkylammonium oleate and linoleate ILs were tested for different metal salts removal	Removal efficiencies of 99% were found for the period IV transition metal salts. No extraction happened for NaCl.	[66]
1-sodium acetate-2,3-dimethyl imidazolium iodide ionic liquid (IL-1) and 1-sodium propionate-2,3-dimethyl imidazolium iodide for the removal of NaCl	High water fluxes and negligible NaCl loss. Efficient salt removal.	[67]

**Table 2 membranes-13-00874-t002:** Overview of the recent research of GO desalination membranes.

**GO and Membranes in Desalination Studies**	**Main Findings**	**Reference**
GO multilayers were coated on polyamide TFC membrane using layer-by-layer deposition method of the oppositely charged GO nanosheets	Increased hydrophilicity and surface roughness which leads to enhanced antifouling NaCl salt rejection up to 96.4% The membrane exhibits improved resistance to chlorine, effectively preventing degradation when exposed to high-salinity water	[33]
Nanofiltration polyethylenimine membrane modified with GO via layer-by-layer assembly used for water softening	High salt rejection, enhanced permeability, and water flux	[116]
Modified GO with 3-(aminopropyl)triethoxysilane (APTS) embedded in PVDF membrane for salt rejection	Enhanced water flux by 86%, excellent salt rejection (>99.9%) Improved surface and bulk porosity. The increased mean pore size led to more hydrophilic interactions attributed to the presence of GO functional groups	[117]
Graphene oxide dispersed polysulfone MMM prepared via wet phase inversion method	Enhanced hydrophilicity and Na_2_SO_4_ rejection with a maximum of 72%	[118]
GO surface-deposited poly(amide-imide)–polyethyleneimine (PAI–PEI) hollow fiber membrane for MgCl_2_, CaCl_2_, and NaCl salt solutions filtration	Improved mechanical properties, selectivity, and 86% higher pure water permeability Enhanced hydrophilicity due to the acidic groups (e.g., carboxylic acid and hydroxyl) added on the surface after incorporating GO to the polymer membrane	[119]
Thin-film composite polyamide membranes surface-functionalized by GO covalently bonded to the surface	Antimicrobial properties enhancement. 97.8% NaCl salt rejection and 41.4 L/m^2^ h water flux.	[120]
Functionalized graphene membrane with limited hydroxyl group addition for NaCl removal from aqueous solutions	Higher water flux, satisfactory salt rejection, and excellent desalination efficiency	[121]
GO deposition in polymeric membranes such as polycarbonate, polyvinylidene fluoride, polysulfone and poly(acrylonitrile)	Achieved the required permeate flux and mechanical strength High pure water permeate and excellent rejection	[122]
Graphene oxide coated with tannic acid (GOT) and incorporated as filler material in polyamide RO membrane	Improved chlorine resistance and antimicrobial properties Increased water flux and salt rejection with increasing GOT wt%	[123]

**Table 3 membranes-13-00874-t003:** Overview of the recent research on hybrid adsorptive desalination membranes.

**Hybrid Membranes**	**Main Findings**	**Reference**
Polyamide RO membrane was modified with choline chloride–urea-based DES and commercial ILs (1-hexyl-3-methyl-imidazolium chloride, 3-methyl-1-octyl-imidazolium tetra fluoroborate, N-butyl pyridinium and betaine monohydrate)	An increase of 27% and 3% in water flux and NaCl rejection was observed N-butyl pyridinium and betaine monohydrate proven to have the best desalination results	[144]
GO membrane was improved with choline chloride- and ethylene glycol-based novel DESs for dye/salt filtration	Four filtration cycles achieved 74–100% flux recovery with rejections of 99.9% for dye and 5.4% for Na_2_SO_4_/NaCl salts Water permeation rates 7.1 times higher than the unmodified GO membrane for the DES/GO functionalized membrane	[145]

## Data Availability

No new data were created or analyzed in this study. Data sharing is not applicable to this article.

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
