# Peer review of "Adsorptive Membranes Incorporating Ionic Liquids (ILs), Deep Eutectic Solvents (DESs) or Graphene Oxide (GO) for Metal Salts Extraction from Aqueous Feed"

_membranes, 2023, doi:10.3390/membranes13110874_

Round 1
Reviewer 1 Report (Previous Reviewer 1)
Comments and Suggestions for Authors
In my opinion, the authors replied to all comments. The article can be accepted after minor revisions.
Please check the guidelines of the journal for the references. Be consistent with the authors' names; write the names of all authors or only the first one, plus et al.
Author Response
Please see the attachment.

Reviewer 2 Report (New Reviewer)
Comments and Suggestions for Authors
The review entitled "Adsorptive Membranes Incorporating Ionic Liquids (ILs), Deep Eutectic Solvents (DESs) or Graphene Oxide (GO) for Metal Salts Extraction from Aqueous Feed" summarises a considerable amount of literature on the use of ILs, DESs and GOs in membrane preparation. The topic is highly topical and interest in the subject is growing rapidly, as well described in the introduction. The examples are well described and the structure of the text is appropriate.
However, the section on ILs and DESs is chaotic. There is often confusion between the properties of one and the other. There is an interesting comparison of the actual properties of ILs and DESs in the recent review by Afonso et al. (10.1039/d2gc03198a). For example, it is not true that DESs have high thermal and chemical stability. Similarly, it is not true that they necessarily have low volatility. As they are mixtures, these properties depend on the chemical nature of their components. The very definition of DES is also imprecise. Martins et al. (10.1007/s10953-018-0793-1) describe well how DES show a strong deviation from the ideal eutectic mixture. Remaining in the DES section, the main applications are not reported as for ILs. DES are widely used as reaction solvents (10.1039/d1gc03714e), in the dissolution of important natural polymers (10.3390/molecules25163652), in polyphenol extraction (10.3390/molecules28196852), in chiral electroanalysis (10.1016/j.electacta.2021 .138189), electrochemistry (10.1016/j.cogsc.2022.100649) and pharmaceutical applications (10.1016/j.ijpharm.2022.121811).
At the same time, there are some inaccuracies in the section on ILs. The definition of ionic liquids based on a melting temperature below 100 °C is outdated (10.1007/s12551-018-0419-2). In addition, some references in the applications section need to be added:
"including catalysis (10.1039/c7nj04707j), polymer dissolution and transformation (10.1039/C6GC02935C), physical chemistry (10.1039/B600519P), polymer science (10.3390/molecules28093836), cell biology (10.1016/j.bbamem.2019.183103), materials science (10.1021/acs.jpclett.3 c00329), electrochemistry (10.3390/molecules25245812), desalination (10.1038/s41467-020-20706-y), energy production (10.1021/acs.chemrev.6b00504), nuclear physics (10.1021/cr200193x), medicinal chemistry (10.1016/j.biotechadv.2022.108055), and various engineering fields."
The full stop from line 227 to 233 should be deleted as it is redundant with the next paragraph.
Minor
A figure on the anion and cation structures of ILs should be added, as for DESs.
The quality of Figure 1 should be improved.
The quality of Figure 2 should be improved. In addition, HBAs such as choline chloride, betaine and proline need to be added.
The overall quality of the tables needs to be improved and made more attractive.
Some figures extracted from the manuscripts of the most important examples should be added. The graphical part is very important in a review.
Comments on the Quality of English LanguageModerate editing of English language required
Round 2
Reviewer 2 Report (New Reviewer)
Comments and Suggestions for Authors
I would like to thank the authors for accepting all my suggestions. The manuscript is now suitable for publication.
This manuscript is a resubmission of an earlier submission. The following is a list of the peer review reports and author responses from that submission.
Round 1
Reviewer 1 Report
Comments and Suggestions for Authors
The manuscript is a review of the extraction of salts from water using Adsorptive Membranes. In my opinion, the review is interesting, but the literature must be improved together with the environmental aspect of ILs and DES.
In the first part of the review, the order of description is GO>ILs>DES. I advise to do the same in the second part during the description of the membranes.
Line 83: “One approach involves the development of nanocomposite or nano enhanced membranes, where nanomaterials like metal oxides (such as zeolites and silica), carbon-based nanotubes, and hydrophilic functional groups are seamlessly integrated into the polymeric membrane matrix.”. Can you add a reference?
Line 156: Ionic Liquids. There is literature about the “side-effects” of ILs in terms of toxicity, for example. I suggest boosting the description of the ILs, also considering this aspect. For example:
https://doi.org/10.1016/j.scitotenv.2021.147309
Line 167: (CH3)3NHCl/2AlCl3, the numbers are in subscript. Please check the whole manuscript.
Lines 170-175: I advise adding literature to support the sentence about the properties and applications of ILs. For example:
https://doi.org/10.1016/j.molliq.2021.117210
Line 219: Regarding the environmental activity and toxicity, I advise also considering the side effects of DES. For example: https://doi.org/10.1016/j.heliyon.2022.e12567
Line 243: change “created” with “prepared”.
Line 245: 70 °C or -70 °C?
Line 306: In my opinion, this part is in contrast with the other section about ILs. They cannot be defined as green solvents if they are not biodegradable with potential toxicity. I strongly advise to read the manuscript again and make the text more coherent. For example: https://doi.org/10.1016/j.rser.2021.112039
Line 439: change “particular” to “another”.
Line 462: “Escherichia coli” in Italics. Please, check the rest of the manuscript.
Line 481: 3.3 Incorporation of deep eutectic solvents in membranes. Are there studies about desalination using DES in membranes? If there is not literature, it cannot be written “has not been extensively explored”.
Line 498: DES are “prepared” and not ”synthesized”. ILs are synthesized. Please, check the entire article.
I advise major revisions, and if the answers of the authors are acceptable, the article could be considered for publication. The work is not recommended for publication in its present form.
Comments on the Quality of English LanguageIn my opinion, the English is fluent.
Reviewer 2 Report
Comments and Suggestions for Authors
I regret to inform you that after careful consideration, I cannot recommend the acceptance of the manuscript titled "Adsorptive Membranes Incorporating Ionic Liquids (ILs), Deep Eutectic Solvents (DESs) or Graphene Oxide (GO) for Metal Salts Extraction from Aqueous Feed " for publication. While the language used in the manuscript is excellent, there are significant issues regarding the clarity and accuracy of the information presented, which make it misleading and scientifically unacceptable.
The primary concern lies in the confusion of different types of membranes and their respective applications within the manuscript. The authors fails to distinguish between membranes used in wastewater treatment, brackish water desalination, softening, and seawater desalination. Consequently, the manuscript lacks clear differentiation and fails to address the unique challenges and requirements associated with each application. This lack of clarity results in a convoluted representation of the subject matter, making it difficult for readers to understand which applications the described membranes are intended for. Such confusion undermines the scientific rigor of the manuscript and hinders its potential to provide valuable insights to the scientific community.
Although I appreciate the linguistic quality of the manuscript, it is imperative that the information presented is scientifically accurate, clear, and focused. I strongly recommend that the author revises the manuscript, placing a clear emphasis on distinguishing between different membrane types and their specific applications. This revision will enhance the scientific integrity and coherence of the work.
In conclusion, I regretfully reject the submitted manuscript due to the confusion caused by mixing different membrane types and their applications. The misleading nature of the manuscript's content undermines its scientific acceptability. I encourage the author to address these concerns and revise the manuscript accordingly.
